# Epithelioid Pleural Mesothelioma Is Characterized by Tertiary Lymphoid Structures in Long Survivors: Results from the MATCH Study

**DOI:** 10.3390/ijms23105786

**Published:** 2022-05-21

**Authors:** Laura Mannarino, Lara Paracchini, Federica Pezzuto, Gheorghe Emilian Olteanu, Laura Moracci, Luca Vedovelli, Irene De Simone, Cristina Bosetti, Monica Lupi, Rosy Amodeo, Alessia Inglesi, Maurizio Callari, Serena Penpa, Roberta Libener, Sara Delfanti, Antonina De Angelis, Alberto Muzio, Paolo Andrea Zucali, Paola Allavena, Giovanni Luca Ceresoli, Sergio Marchini, Fiorella Calabrese, Maurizio D’Incalci, Federica Grosso

**Affiliations:** 1Department of Biomedical Sciences, Humanitas University, Via Rita Levi Montalcini 4, Pieve Emanuele, 20072 Milan, Italy; lara.paracchini@humanitasresearch.it (L.P.); rosy.amodeo@humanitasresearch.it (R.A.); paolo.zucali@hunimed.eu (P.A.Z.); maurizio.dincalci@hunimed.eu (M.D.); 2Laboratory of Cancer Pharmacology, IRCCS Humanitas Research Hospital, Via Manzoni 56, Rozzano, 20089 Milan, Italy; monica.lupi@humanitasresearch.it (M.L.); sergio.marchini@humanitasresearch.it (S.M.); 3Department of Cardiac, Thoracic, Vascular Sciences and Public Health, University of Padova Medical School, 35128 Padova, Italy; federica.pezzuto@unipd.it (F.P.); laura-moracci@libero.it (L.M.); luca.vedovelli@ubep.unipd.it (L.V.); fiorella.calabrese@unipd.it (F.C.); 4Laboratorul de Anatomie Patologică, Spitalul Clinic de Boli Infecțioase și Pneumoftiziologie Victor Babeș, 300223 Timisoara, Romania; olteanugheorgheemilian@gmail.com; 5Department of Oncology, Istituto di Ricerche Farmacologiche Mario Negri IRCCS, Via Mario Negri 2, 20156 Milan, Italy; irene.desimone@marionegri.it (I.D.S.); cristina.bosetti@marionegri.it (C.B.); alessia.inglesi@unimi.it (A.I.); 6Michelangelo Foundation, 20121 Milan, Italy; maurizio.callari@fondazionemichelangelo.org; 7Dipartimento Attività Integrate Ricerca e Innovazione (DAIRI), Infrastruttura Ricerca Formazione Innovazione (IRFI), Azienda Ospedaliera SS Antonio e Biagio e Cesare Arrigo, 15121 Alessandria, Italy; serena.penpa@esterni.ospedale.al.it (S.P.); rlibener@ospedale.al.it (R.L.); 8Mesothelioma Unit, Azienda Ospedaliera SS Antonio e Biagio e Cesare Arrigo, 15121 Alessandria, Italy; sara.delfanti@ospedale.al.it (S.D.); antonina.deangelis@ospedale.al.it (A.D.A.); 9SC Oncologia, Ospedale Santo Spirito, 15033 Casale Monferrato, Italy; amuzio@aslal.it; 10Department of Oncology, IRCCS Humanitas Research Hospital, Rozzano, 20089 Milan, Italy; 11Department Immunology, IRCCS Humanitas Clinical and Research Center, Rozzano, 20089 Milan, Italy; paola.allavena@hunimed.eu; 12Department of Medical Oncology, Saronno Hospital, ASST Valle Olona, Saronno, 21047 Varese, Italy; giovanniluca.ceresoli@gmail.com

**Keywords:** mesothelioma, tertiary lymphoid structures, long survivors, transcriptomics, B cells, CD20

## Abstract

Pleural mesothelioma (PM) is an aggressive tumor with few therapeutic options. Although patients with epithelioid PM (ePM) survive longer than non-epithelioid PM (non-ePM), heterogeneity of tumor response in ePM is observed. The role of the tumor immune microenvironment (TIME) in the development and progression of PM is currently considered a promising biomarker. A few studies have used high-throughput technologies correlated with TIME evaluation and morphologic and clinical data. This study aimed to identify different morphological, immunohistochemical, and transcriptional profiles that could potentially predict the outcome. A retrospective multicenter cohort of 129 chemonaive PM patients was recruited. Tissue slides were reviewed by dedicated pathologists for histotype classification and immunophenotype of tumor-infiltrating lymphocytes (TILs) and lymphoid aggregates or tertiary lymphoid structures (TLS). ePM (*n* = 99) survivors were further classified into long (>36 months) or short (<12 months) survivors. RNAseq was performed on a subset of 69 samples. Distinct transcriptional profiling in long and short ePM survivors was found. An inflammatory background with a higher number of B lymphocytes and a prevalence of TLS formations were detected in long compared to short ePM survivors. These results suggest that B cell infiltration could be important in modulating disease aggressiveness, opening a pathway for novel immunotherapeutic approaches.

## 1. Introduction

Pleural mesothelioma (PM) is a rare malignancy of the pleural lining mainly caused by inhaled asbestos. The difficulty in phagocytizing these mineral fibers leads to the onset of a chronic immune response [1]. For this reason, the strong interaction between PM tumor cells and the surrounding tumor immune microenvironment (TIME) represents a key to the comprehension of tumor behavior and the basis for the development of pharmacological treatments.

The TIME is composed of several actors that strictly interact with each other. B cells play a fundamental role in the adaptive immune response. Recently, they have attracted particular attention because they are involved in the formation of lymphoid ensembles called tertiary lymphoid structures (TLS) [2,3,4]. TLS are ectopic lymphoid aggregates of immune cells, mainly B and T cells, that typically emerge in chronically inflamed environments like autoimmune diseases, where they are believed to sustain the aberrant immune response to autoantigens expressed in the target organs. TLS have been described in several cancer types; their role in the antitumor response is not completely understood, but many studies reported TLS as a positive prognostic and predictive factor in solid tumors [2,5,6,7].

Histology is a well-known prognostic factor in PM, with epithelioid PM (ePM) displaying longer survival than the non-epithelioid (non-ePM) [8]. However, very heterogenous outcomes have been registered also in the ePM histotype. To date, biological or molecular features that can predict different outcomes have not yet been identified. A recent study has suggested the CTGF protein as a prognostic factor for ePM [9]; instead, others focused on immune-related proteins such as the V-domain Ig suppressor of T cell activation (VISTA), which is highly expressed in ePM, but less in those patients with aggressive tumors [10,11]. A focus on the TIME and, in particular, on the role of B cells in PM has not been investigated so far. Few therapeutic options are available for PM, and they mainly rely on chemotherapy with a median overall survival of 14 months. Progress has been achieved with immunotherapy mainly in the non-ePM subtype as sustained by the CheckMate 743 trial [12]. However, novel paths can be pursued.

In this scenario, our work aimed to identify TIME features in a retrospective cohort of chemonaive PM patients that could explain patients’ different outcomes and guide hypothesis generation for novel therapeutic approaches.

## 2. Results

### 2.1. Cohort Description

A total of 186 patients affected by pleural mesothelioma (PM) were included in this study, following strict inclusion criteria (Appendix A). The final selected cohort encountered 129 patients (Appendix A), composed of 32 women (24.8%) and 97 men (75.2%), with a median age at the time of diagnosis of 77 years (range 34–97). The main clinical data are summarized in Table 1.

### 2.2. Pathological Examination

Two dedicated pathologists reviewed 130 samples of PM (one patient had two samples) for comprehensive histological examination. Of 129 tumors, 99 (77%) were classified as ePM and 30 (23%) as non-ePM, including both sarcomatoid and biphasic subtypes.

### 2.3. ePM and Non-ePM Have Different Transcriptional Signatures

After the evaluation of quality standards, 69 samples (one patient has two samples)—45 (66%) ePM and 23 (34%) non-ePM—were selected for molecular analyses (Appendix A). Unsupervised analysis of the whole transcriptome of PM tumors showed that ePM (in blue) and non-ePM (in red) formed two clusters, though with low variance (17% on the PC1 and 10% on the PC2) (Figure 1). To identify their transcriptional profile, differential expression analysis was performed. In the analysis, 817 differentially expressed genes (DEGs) were found (Appendix A), mainly involved in immune-regulatory processes (immune system, cytokine and interferon signaling, major histocompatibility complex pathways), preferentially activated in the non-ePM subtype (Appendix A).

### 2.4. Long Survivor ePM Tumor Microenvironment Presents a Higher Fraction of B Cells Than Short Survivor ePM

Focusing on ePM, a comparison between long and short survivors was performed, showing no significant differences in the transcriptional features of the two groups. A deconvolution approach through quanTISeq [13] was also implemented, allowing the definition of a percentage of immune infiltrating cells, i.e., B cells, M1 and M2 macrophages, monocytes, neutrophils, natural killer cells, non-regulatory CD4+ T cells, CD8+ T cells, regulatory CD4+ T cells, and dendritic cells. Interestingly, long survivor ePM samples showed overexpression of B cells compared with short survivors, which had a prevalence of neutrophils and M2 macrophages (Figure 2).

### 2.5. Long Survivor ePM Tumors Are Characterized by Tertiary Lymphoid Structures

Driven by the results obtained from quanTISeq, ePM samples of each cluster (24 of Group A and 12 of Group B) were selected and histologically revised, focusing on tumor-infiltrating lymphocytes (TILs). Both B (CD20+) and T (CD3+) lymphocytes were evaluated in a semiquantitative way (scoring from 0 to 3) together with the presence of lymphoid aggregates and/or TLS (Figure 3). Interestingly, the high number of B cells in Group B was confirmed by immunohistochemistry (IHC) (Figure 4). B (CD20+) IHC scores were higher in long survivor than short survivor ePM (*p* = 0.025). In multivariate analysis, the prevalence of B (CD20+) lymphocytes in long survivor PMs was confirmed (Log Odds Ratio = −0.06; 95% Confidence Interval = −0.12,−0.001; *p* = 0.039).

Moreover, morphological assessment of lymphoid aggregates revealed that B cell prevalence in Group B matched TLS formation (83% in Group B and 37% in Group A, respectively). This was in line with the assessment of *CXCL13*, a gene encoding for a chemokine associated with TLS, and *MS4A1*, encoding for CD20 marker which was highly expressed in long survivors in transcriptomic analysis (Figure 3).

## 3. Discussion

In this retrospective series of PM, we found that long ePM survivors (>36 months) had a specific inflammatory background with a higher number of B lymphocytes and a prevalence of TLS formations compared to short survivor ePM. To the best of our knowledge, no other previously published study has focused on B cells and TLS in PM. The exact role played by B lymphocytes in the immune system–tumor interplay is unclear. However, recent progress in the detailed description of B lymphocytes in the TIME shows them to be central to the immune crosstalk. Briefly, B cells in a TIME can play a positive role (i.e., positive regulation of cancer). In case of negative modulation of tumor growth, such as in ovarian cancer, non-small cell lung carcinoma, and cervical cancer, the increased presence of CD20+ B-cell TILs correlated with improved survival and subsequent lower relapse rates. The probable mechanism behind the negative regulation of cancer by B cells can be explained by the secretion of cytokines with effector functions that would promote T cell polarization towards a Th1 or Th2 phenotype. Another probable mechanism can be directly connected with their role as antigen-presenting cells (via neoantigens), thus promoting an antitumor response mediated by T cells in the TIME [14]. The positive regulation of cancer by B cells in the TIME has been directly linked to B cells that have signal transducer and activator of transcription 3 (STAT3) activated, which eventually promotes tumor growth by contributing to a proangiogenic environment [15]. Following this positive regulation of tumor growth, several other works have reported a correlation between tumors with TIMEs rich in CD20+ and CD138+ B cells and poor survival.

In our study, high numbers of CD20+ B cells were preferentially expressed in ePM tumors of long survivors. At this time, there are no mechanistic studies that explored this intriguing finding. As reported in other studies, a plausible explanation could be related to the role played by B cells in their antigen-presenting ability and priming of cytolytic antitumor T cell responses [14]. The presence of neoantigens is sometimes viewed as synonymous with tumor mutational burden. However, unlike many carcinogen-related tumors, PM shows a low mutational burden, resulting in a low number of neoantigens. Nevertheless, there is evidence that chromosomal rearrangements through the mechanism of chromothripsis can trigger the formation of novel fusion genes and neoantigens that can be the nidus for B-cell-related antitumor immunity, specifically in PM [16,17].

Regarding the TLS presence in the long ePM survivor group, this is in line with other published studies about various malignancies wherein the presence of TLS correlates with a better outcome [3,18]. A role for TLS in priming the local immune response and in lymphocyte recruitment has been suggested; indeed, the presence of TLS in tumors is usually associated with increased infiltration of T lymphocytes [6,7,18].

A few studies have evaluated the presence of TLS in epithelioid malignant peritoneal mesotheliomas (EMPeM). Benzerdjeb et al. [19] demonstrated that TLS in EMPeM was significantly associated with neoadjuvant chemotherapy, denoting a post-treatment change to the TIME, but not to overall survival or progression-free survival, thus lacking a prognostic significance.

A previous study that analyzed a large number of ePM [20], albeit only using IHC, showed that the ratio of M2 polarized macrophages to CD8+ or CD20+ TILs was an independent predictor of survival of ePM. Indeed, low numbers of M2-macrophages and a high number of CD20+ B cells were statistically significant prognostic factors in these patients. Our analysis shows that group B of epithelioid long survivors (>36 months) had high fractions of B cells, which showed overlap with high numbers of CD20+ B cells highlighted by IHC and low fractions for M2 polarized macrophages (transcriptional) which were not evaluated through IHC but were not observed in the pathology morphological assessment. Our results confirm the data present in the literature regarding the IHC evaluation of TIMEs in ePM. Moreover, they provide transcriptional evidence about the population of immune cells that is involved in the tumor–immune system crosstalk, specifically those that can predict improved survival for ePM patients [20].

Our study has some limitations. First, it is retrospective in nature; however, the observations made will be confirmed in the ongoing prospective part of the study, which has already concluded the enrollment. Second, the number of cases included is relatively low; however, the multicentric recruitment made it possible to obtain a relatively high number of patients for such a rare neoplasm. The third limitation regards the immunophenotypic characterization that was only possible for few TIME markers. Nonetheless, we were able to exclude the presence of other components—for example, macrophages—through a detailed morphological analysis. Furthermore, we focused on the populations that are probably the most represented and with a certain pilot role in this context and for which there are still many knowledge gaps.

In conclusion, our study suggests that B (CD20+) lymphocytes could influence the behavior of the disease and emphasizes a possible new path to be explored in translational studies that use B cells for immunotherapy. PM is indeed a very difficult-to-treat disease with limited treatment options despite recent approaches modulating the tumor microenvironment having shown improvements in survival. The addition of antiangiogenic agents to standard chemotherapy extended overall survival both in first- and in second-line chemotherapy [21,22], and more recently, the association of ipilimumab and nivolumab (CheckMate 743) significantly prolonged overall survival, especially in the non-ePM subtype [12]. A larger number of PM cases and mechanistic studies is required to validate and understand B cell functioning in this context. The use of morphological assessment for the presence of TLS and its use as a potential prognostic marker should be evaluated in prospective studies. Moreover, it could provide new insight into the role of B cells in PM TIME that could potentially be exploited as new targets for interventions.

## 4. Materials and Methods

### 4.1. Patient Selection

A retrospective cohort of 186 patients was selected from seven Italian hospitals from 2009 to 2019. The selection of cases was performed according to the following criteria: (i) availability of formalin-fixed, paraffin-embedded (FFPE) tumor samples, (ii) presence of the anatomopathological classification of epithelioid PM (ePM) and non-epithelioid PM (non-ePM), and (iii) availability of survival data to distinguish ePM long (overall survival > 36 months) and ePM short (overall survival < 12 months) survivors. All tumor samples were collected during the diagnostic pleuroscopic/thoracoscopic procedures. Signed informed consent was obtained from each patient. Only the unique identifier (ID) of each patient was transmitted to the central laboratory; no clinical data were shared with pathologists. The study was conducted in accordance with the principles of the Declaration of Helsinki and approved by the local scientific ethical committees.

### 4.2. Tumor RNA Isolation and Libraries Preparation

After scraping the histologic slides where the FFPE samples were fixed, tumor RNA was semi-automatically (QIAcube, Qiagen, Hilden, Germany) extracted and purified with miRNeasy FFPE kit (Qiagen, Hilden, Germany) following the manufacturer’s procedures. The quality of RNA extracted was evaluated (TapeStation 4200, Agilent Technologies, Santa Clara, CA, USA) and its amount was quantified using the fluorimetric method Qubit RNA BR (Invitrogen, Carlsbad, CA, USA). Libraries for transcriptomic analysis were prepared using the TruSeq Stranded Total RNA kit (Illumina, San Diego, CA, USA) starting from 100 ng of total RNA. Following the manufacturer’s instruction, after the conversion of RNA in a complementary double-strand cDNA, ends were blunted and 3′-end adenylated. The following barcoding procedure allowed the pooling and sequencing of libraries obtained on the NextSeq-500 platform (Illumina, San Diego, CA, USA), ensuring at least 30 million reads per sample.

### 4.3. High Throughput Sequencing Data Analysis

Raw data sequences were demultiplexed with bcl2fastq Conversion Software (Illumina) using a no-lane-splitting parameter. Two reads, R1 and R2, were obtained per sample. Quality control was performed with FastQC [23]. Quality data were visualized through MultiQC [24]. Data analysis was performed with the publicly available pipeline bcbio-nextgen [25]. fastq files were aligned with the hisat2 aligner [26] using the hg38 version of the human genome. Post-alignment quality control was performed with bcbioRNASeq [27]. Gene counts were computed with Salmon [28]. Salmon quantification was read with tximport [29] and differential expression analysis was performed with DESeq2 package [30]. Counts were filtered considering at least 10 reads per gene. Unsupervised classification analysis with principal component analysis was performed with DESeq2 package [30]. Two differential expression analyses were done: (i) ePM class was compared versus non-ePM class; (ii) long ePM survivor class was compared versus short ePM survivors. Differentially expressed genes (DEGs) were considered with a *p*-adjust less than 0.05 (multiple testing correction with false discovery rate). DEGs were associated with pathways through gene set enrichment analysis (GSEA) [31], sorting genes according to their log2 Fold Change from the most upregulated to the most downregulated, using the Reactome database [32]. GSEA was performed with the cluster profile package [33]. Normalized log values of gene counts were converted into z-score for data visualization and clustering. Clustering was performed with the Ward variance minimization algorithm.

### 4.4. Computational Analysis of the Tumor Microenvironment

Analysis of the tumor microenvironment was performed with a deconvolution approach called quanTIseq [13]. In detail, the quanTIseq pipeline was used from bash using the --tumor = TRUE setting starting from transcript per million (TPM) data. Through this method, the following components of the tumor microenvironment were considered: B cells, M1 and M2 macrophages, monocytes, neutrophils, natural killer cells, nonregulatory CD4+ T cells, CD8+ T cells, regulatory CD4+ T cells, and dendritic cells. The “other uncharacterized cells” represent a proxy of the tumor component. For data visualization and analysis, the categories with low variance across the cohort were removed. Only the most informative components, such as nonregulatory CD4+ T cells, CD8+ T cells, B cells, and neutrophils M1 and M2 macrophages, were considered.

Clustering was performed with the Ward variance minimization algorithm.

### 4.5. Histological and Immunohistochemical Evaluation

Two dedicated pathologists reviewed 130 slides for histological examination. Each case was defined as adequate when larger than 2 cm^2^ and/or with 60% of neoplastic cells, as previously described [34]. All histological evaluations were performed independently. Discordant cases were reviewed and discussed to achieve a final diagnosis. Each tumor was classified into histotypes (epithelioid, biphasic, and sarcomatoid), according to the 2021 WHO classification [35]. Additionally, fibrosis, necrosis, and inflammation were morphologically evaluated over the entire surface and expressed as percentages. The presence of lymphocyte aggregates and/or follicles (i.e., TLS) was also recorded. Inflammatory cells were further classified into lymphocytes (B and T) by using the following primary antibodies: anti-CD20 (1:200, Dako, clone L26, CD20CY), anti-CD3 (1:200, Leica, clone NCL-L-CD3-565). Immunoreactivity was quantified with a score 0–3 (0: absent; 1:<10%; 2:10–30%; 3:>30%) as previously described [1]. Immunohistochemical analyses were performed by using the Bond automated system (Leica Bond III, Leica Microsystems Srl, Wetzlar, Germany).

### 4.6. Statistical Analyses

Continuous data were reported as median (interquartile ranges); categorical data were reported as percentage and absolute frequencies. Wilcoxon rank-sum tests were performed for continuous variables and the Fisher’s exact test or Pearson chi-square test for categorical variables. A logistic mixed model (estimated using ML and Nelder–Mead optimizer) with random intercepts on each individual patient and age, sex, systemic treatment, surgery, radiotherapy, prevalent pattern, inflammation percentage, CD20+ percentage, and CD3+ percentage as covariates was fitted to predict long survivor and short survivor ePM. Statistical significance was set at *p* < 0.05.

## Figures and Tables

**Figure 1 ijms-23-05786-f001:**
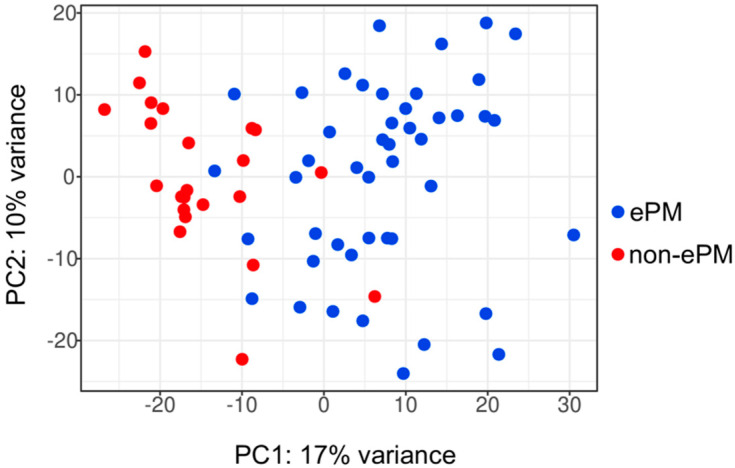
Principal component analysis (PCA) of the RNA-Seq cohort of samples. In blue ePM, epithelioid pleural mesothelioma; in red non-ePM, non-epithelioid pleural mesothelioma. The *x*-axis indicates principal component 1; the *y*-axis indicates principal component 2.

**Figure 2 ijms-23-05786-f002:**
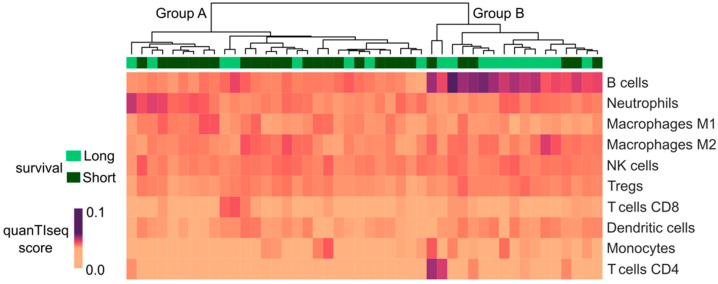
Evaluation of immune infiltrating cells through the quanTIseq deconvolution approach. From the top downwards: unsupervised clustering of the ePM cohort divided into Group A and Group B. The green line represents survival: long survivors in light green; short survivors in dark green. The heatmap shows the quanTIseq score through a range of colors as in the legend: the darker the color, the higher the score.

**Figure 3 ijms-23-05786-f003:**
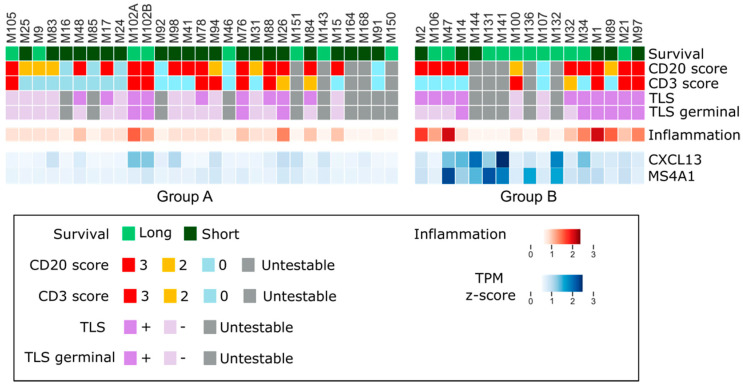
Immunostaining and transcriptomic overlap. From the top downwards: the two groups of the ePM, A and B, are reported—long survivors in light green and short survivors in dark green. CD20 and CD3 scores as in the legend. TLS and TLS germinal as in the legend. Untestable cases are indicated in grey. Inflammation percentage is reported in normalized z-score. *CXCL13* and *MS4A1* are reported with their TPM z-score.

**Figure 4 ijms-23-05786-f004:**
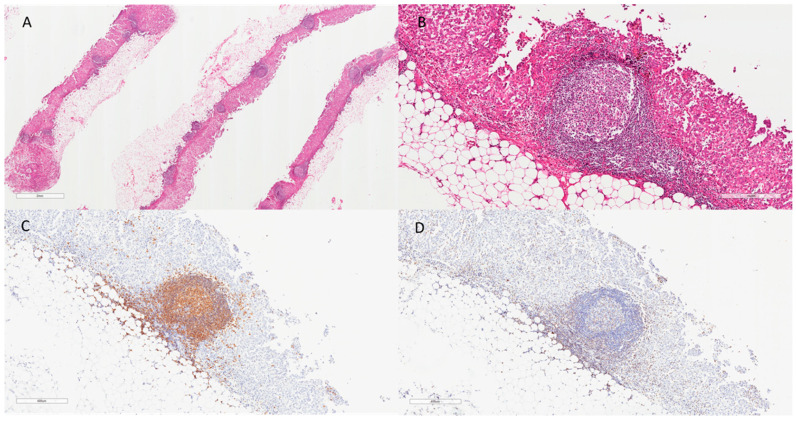
Figure panel reporting an emblematic case (M34) of an ePM with a high number of lymphoid aggregates and TLS ((**A**), hematoxylin and eosin, scale bar: 2 mm), with prominent germinal centers ((**B**), hematoxylin and eosin, scale bar: 300 µm). A high number of CD20 (**C**) and a lower number of CD3 (**D**) were also detected (IHC, scale bar: 400 µm).

**Table 1 ijms-23-05786-t001:** Clinical data of the pleural mesothelioma (PM) cohort.

		Cases	
	All Patients	Long Survival	Short Survival
	*n* (%)	*n* (%)	*n* (%)
Cohort	129	45	54
Median age (years), range	77 (34–97)	73 (52–91)	82 (34–97)
Gender (%)			
Male	97 (75.2)	36 (80.0)	34 (63.0)
Female	32 (24.8)	9 (20.0)	20 (37.0)
Histologic subtype			
Epithelioid (ePM)	99 (77)	45 (100.0)	54 (100.0)
Non-epithelioid (non-ePM)	30 (23)	0 (0.0)	0 (0.0)
Survival classification ^1^			
Long survival (>36 months)	45 (34.9)	45 (100.0)	0 (0.0)
Short survival (<12 months)	54 (41.9)	0 (0.0)	54 (100.0)
Other	30 (23.3)	0 (0.0)	0 (0.0)
ECOG performance status			
0	49 (38.0)	24 (53.3)	16 (29.6)
1	12 (9.3)	6 (13.3)	4 (7.4)
2	5 (3.9)	0 (0.0)	2 (3.7)
3	1 (0.8)	0 (0.0)	0 (0.0)
Unknown	62 (48.1)	15 (33.3)	32 (59.3)
First line treatment			
No	22 (17.1)	0	10 (18.5)
Carboplatin + pemetrexed	22 (17.1)	7 (15.6)	8 (14.8)
Cisplatin + pemetrexed	6 (4.7)	5 (11.1)	1 (1.9)
Platin derivate + pemetrexed + Additional drug	4 (3.1)	1 (2.2)	3 (5.6)
Treatment combination not containing platinum + pemetrexed	5 (3.9)	1 (2.2)	2 (3.7)
Unknown	70 (54.3)	31 (68.9)	30 (55.6)
Surgery			
Yes	19 (14.7)	18 (40.0)	1 (1.9)
No	109 (84.5)	26 (57.8)	53 (98.2)
Unknown	1 (0.8)	1 (2.2)	0 (0.0)
Radiotherapy			
Yes	10 (7.8)	10 (22.2)	0 (0.0)
No	117 (90.7)	33 (73.3)	54 (100.0)
Unknown	2 (1.6)	2 (4.4)	0 (0.0)

^1^ Only for ePM subtype.

## Data Availability

Raw sequencing data for this study have been deposited in the European Nucleotide Archive (ENA) at EMBL-EBI under accession number PRJEB52839. Raw counts RNASeq data are available at Zenodo, https://doi.org/10.5281/zenodo.6532036 accessed on 9 May 2022.

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
