# Peer review of "Epithelioid Pleural Mesothelioma Is Characterized by Tertiary Lymphoid Structures in Long Survivors: Results from the MATCH Study"

_ijms, 2022, doi:10.3390/ijms23105786_

Round 1

Reviewer 1 Report

Overall Evaluation:Thanks for giving me to review this manuscript. Authors investigated TIME features in a retrospective cohort of chemonaive PM patients that could explain patients' different outcomes.This is a well-written manuscript. However, there are several  methodological aspects that warrant further clarification and alternative statistical analyses may be indicated.
AbstractAuthors mainly analysed epitheloid PM, so authors should show the number of ePM.
MajorAuthors wrote in the Background as "eventually rule the therapeutic choice", however, their results did not contain the therapeutic choice.Authors should delete therapeutic choice in the conclusion.
-Table 1Authors should describe some important prognostic factors, including performance status, stage, and treatments, especially in dividing between long and short survivors.

2.3. ePM and non-ePM have different transcriptional signaturesHow did authors conduct unsupervised classification? Scientific articles should have a minimum level of reproducibility.
Or please delete, because they did not mention this in the Discussion.
2.5. Long survivor ePM tumors are characterized by tertiary lymphoid structures   
As mentioned in the comment to table 1, authors should account for confounding factors. I understand that the sample size is small, but the authors should also present the results of the multivariate analysis at the same time.

Author Response

Response to Reviewer 1 Comments

Point 1: Abstract Authors mainly analysed epitheloid PM, so authors should show the number of ePM.

Response 1: We thank the Reviewer for the suggestion. We added the number of ePM in the abstract "(N=99)", page 1, line 49.

Point 2: Major: Authors wrote in the Background as "eventually rule the therapeutic choice", however, their results did not contain the therapeutic choice. Authors should delete therapeutic choice in the conclusion.

Response 2: We thank the Reviewer for this suggestion: we rephrased the sentence as “guide hypothesis generation for novel therapeutic approaches”, page 2, lines 88-89.

Point 3: Table 1: Authors should describe some important prognostic factors, including performance status, stage, and treatments, especially in dividing between long and short survivors.

Response 3: We made a new Table 1 adding the information required: Table 1, pages 2-3.

Point 4: 2.3. ePM and non-ePM have different transcriptional signatures. How did authors conduct unsupervised classification? Scientific articles should have a minimum level of reproducibility.

Or please delete, because they did not mention this in the Discussion.

Response 4: We thank the Reviewer for highlighting this point. We have detailed the Methods section as "Unsupervised classification analysis with Principal Component Analysis was done with DESeq2 package30. Two differential expression analyses were done: i. ePM class was compared versus non-ePM class; ii. long ePM survivors class was compared versus short ePM survivors.", page 8, lines 350-353.

Point 5: 2.5. Long survivor ePM tumors are characterized by tertiary lymphoid structures  

As mentioned in the comment to table 1, authors should account for confounding factors. I understand that the sample size is small, but the authors should also present the results of the multivariate analysis at the same time.

Response 5: We thank the Reviewer for his/her suggestion. We rebuilt the multilevel model adding age, sex, systemic treatments, surgery, and radiotherapy on the model. CD20+ remains a significant (p = 0.039) predictor of long and short survivor status. We added these new model data to the manuscript in the Result, page 5, line 183, and Methods sections, page 9, lines 454-455.

Reviewer 2 Report

Manuscript is well done in all its sections.

I have just one comment on the materials part: authors say that they collected tumor samples during the diagnostic procedures and that signed informed consent were obtained from all participants.

They could  include a frase on the adopted procedures to ensure anonymity and prevent the identification of specific persons involved in the study

 I found some typos in the text; for example, page 6, line 188.  Therefore, the authors should carefully check the text in the final version of the manuscript.

Author Response

Response to Reviewer 2 Comments

Point 1: I have just one comment on the materials part: authors say that they collected tumor samples during the diagnostic procedures and that signed informed consent were obtained from all participants.They could  include a frase on the adopted procedures to ensure anonymity and prevent the identification of specific persons involved in the study

Response 1: We thank the reviewer for this important point. We added this in the Methods section as “Only the unique identifier (ID) of each patient was transmitted to the central laboratory, no clinical data were shared with pathologists.”, page 7, lines 313-314.

Point 2: I found some typos in the text; for example, page 6, line 188.  Therefore, the authors should carefully check the text in the final version of the manuscript.

Response 2: We thank the reviewer for this revision. We have revised the language of the manuscript.

Round 2

Reviewer 1 Report

Authors responded most of my comments, however, I have several additional comments to improve this manuscript.

-Table 1

Authors should include column name.

In the previous comments, I mentioned TNM stage.

This will also be a confounding factor to adjust.

Author Response

Response to Reviewer 1 Comments

Point 1: -Table 1 Authors should include column name.

Response 1: Done in this last revision: Table 1, pages 2-3, we added the header "Cases". Then, all the other columns are reported in bold.

Point 2: In the previous comments, I mentioned TNM stage. This will also be a confounding factor to adjust.

Response 2: Due to the retrospective nature of this study, we did not revise the radiological images of the patients, thus the information on the stage is lacking. Given the well known difficulties in correctly defining the stage in mesothelioma, we decided to split patients between resected and non resected: this variable was included in the multivariate analysis of the last revision (surgery, Yes/No). To note, the stage information will be considered for the ongoing prospective cohort of this study.

Round 3

Reviewer 1 Report

Authors answered all my questions.